# Sex and Cross-Sex Testosterone Treatment Alters Gamma-Hydroxybutyrate Acid Toxicokinetics and Toxicodynamics in Rats

**DOI:** 10.3390/pharmaceutics16010143

**Published:** 2024-01-21

**Authors:** Qing Zhang, Hao Wei, Annie Lee, Melanie A. Felmlee

**Affiliations:** 1Department of Pharmaceutics and Medicinal Chemistry, Thomas J. Long School of Pharmacy, University of the Pacific, Stockton, CA 95211, USA; q_zhang6@u.pacific.edu (Q.Z.);; 2QPS LLC, Newark, DE 19711, USA

**Keywords:** gamma-hydroxybutyrate, toxicokinetics, pharmacokinetics, pharmacodynamics, monocarboxylate transporters, transporter inhibition

## Abstract

Γ-hydroxybutyric acid (GHB) is widely abused due to its sedative/hypnotic and euphoric effects. In recent years, GHB use has witnessed a notable rise within the LGBTQ+ community. GHB is a substrate of monocarboxylate transporters (MCTs) and exhibits nonlinear toxicokinetics, characterized by saturable metabolism, absorption, and renal reabsorption. This study investigates the impact of exogenous testosterone administration on GHB toxicokinetics and toxicodynamics, exploring the potential of MCT1 inhibition as a strategy to counteract toxicity. Ovariectomized (OVX) females and castrated (CST) male Sprague Dawley rats were treated with testosterone or placebo for 21 days. GHB was administered at two doses (1000 mg/kg or 1500 mg/kg i.v.), and the MCT1 inhibitor AR-C 155858 (1 mg/kg i.v.) was administered 5 min after GHB (1500 mg/kg i.v.) administration. Plasma and urine were collected up to 8 h post-dose, and GHB concentrations were quantified via a validated LC/MS/MS assay. Sleep time (sedative/hypnotic effect) was utilized as the toxicodynamic endpoint. Testosterone treatment significantly affected GHB toxicokinetics and toxicodynamics. Testosterone-treated CST rats exhibited significantly lower renal clearance, higher AUC, and increased sedative effect, while testosterone-treated OVX rats demonstrated higher metabolic clearance. AR-C 155858 treatment led to an increase in GHB renal and total clearance together with an improvement in sedative/hypnotic effect. In conclusion, exogenous testosterone treatment induces significant alterations in GHB toxicokinetics and toxicodynamics, and MCT inhibition can serve as a potential therapeutic strategy for GHB overdose in both cisgender and transgender male populations.

## 1. Introduction

Γ-hydroxybutyric acid (GHB) is an endogenous short-chain fatty acid in mammalian tissues [1]. The salt form of GHB, sodium oxybate (Xyrem^®^), was approved for the treatment of narcolepsy associated with cataplexy in adults by the US Food and Drug Administration (FDA) in 2002 [1]. However, the therapeutic purpose of GHB has been overshadowed by its abuse potential. GHB is a frequently abused substance, prominently utilized in the context of raves and drug-facilitated sexual assault, owing to its euphoric, aphrodisiac, and sedative properties [2,3]. The abuse of GHB results in several severe adverse effects, such as drowsiness, coma, bradycardia, respiratory depression, and even fatality [4]. The incidence of GHB-associated fatalities has exhibited a persistent rise since the 1990s, across the United States, the United Kingdom, Western Europe, and Australia [5]. Over the past decade, the utilization of GHB has witnessed a surge within the LBGTQ community, primarily attributed to the occurrence of chemsex [6]. In 2018, approximately 20% of gay and bisexual Australian men reported lifetime GHB use, and 5.4% reported use in the last six months [5]. In the United Kingdom, 22% of gay and bisexual men presenting to treatment for drug addiction reported problematic use of the prodrug of GHB during the period of 2015 to 2016 [7]. A systematic review indicated that men in their late twenties and early thirties are overrepresented in GHB use (55% to 90%) [8]. They are also more predominant in presenting to emergency departments (50–89%) and involved in GHB-related mortality (69–100%) [8]. Between 2012 and 2019, ambulance attendances related to GHB toxicity issues witnessed a substantial surge of 147% in Australia [5]. Currently, there are no clinically existing therapeutic strategies for GHB overdose, and treatment is limited to supportive care.

GHB exhibits limited lipid permeability attributed to its ionization state under physiological pH conditions (pKa~4.7). Monocarboxylate transporters (MCTs; SLC16) have been identified as the primary determinants governing the transport and disposition of GHB [9,10,11]. The MCT family contains a total of 14 members, among which MCT1, MCT2, and MCT4 have been demonstrated as facilitators of GHB transport [9,12]. Additionally, sodium-coupled MCTs (SMCT1 and SMCT2) are involved in the transport of GHB [13]. MCTs and SMCTs are expressed in the basolateral and apical membrane of renal proximal tubules, respectively [14,15]. Alterations in expression levels of renal MCT/SMCT can lead to changes in transport capacity and, therefore, have an impact on GHB renal clearance through alterations in renal reabsorption [16]. Literature evidence indicates that MCTs exhibit sex-specific differences, thereby influencing the plasma concentration of MCT substrates in both human and rodents. Testosterone treatment in rodents can increase MCT1 and MCT4 protein levels in skeletal muscle, which results in an increased efficiency of lactate transport [17]. The effect of testosterone was tissue-specific, as MCT1 expression was not altered in the heart [17]. In humans, men tend to have higher blood L-lactate concentrations during weight training compared to women, possibly due to greater transporter capacity in the muscle [18]. In a study with valproic acid, one of the substrates of MCTs, women showed higher peak plasma concentrations in comparison to men unless they were on hormonal birth control [19], which may be attributed to the sex differences in MCT expression. Previously, our laboratory demonstrated that hepatic MCT and renal MCT/SMCT expression levels are different between sexes and vary over the estrus cycle in female rats in a tissue-specific manner [20,21]. The sex-dependent variations in renal MCTs/SMCTs lead to alterations in GHB toxicokinetics and renal clearance [21]. Moreover, there were significant alterations in renal MCT expression following exogenous sex and cross-sex hormone treatment in ovariectomized (OVX) and castrated (CST) rats [22]. However, there is a paucity of information regarding the influence of sex and cross-sex hormone treatment on GHB toxicokinetics and toxicodynamics. 

GHB exhibits nonlinear toxicokinetics with capacity-limited absorption, metabolism, tissue distribution, and renal elimination [2,23]. Prior investigations have elucidated that metabolism of GHB is the primary pathway of elimination at lower doses, with negligible renal excretion of unchanged drugs [23,24]. At supratherapeutic doses of GHB, metabolism becomes saturated, leading to renal clearance becoming the predominant elimination pathway. Inhibition of renal reabsorption via MCT inhibition has been evaluated as a potential treatment for GHB overdose [10,25,26], among which AR-C 155858 is a potent MCT1 and MCT2 inhibitor with Ki values of approximately 2.3 nM and 10 nM, respectively, while displaying no inhibitory effect on MCT4 [27]. Prior literature demonstrated the efficacy of AR-C 155858 in improving GHB toxicokinetics and toxicodynamics through the inhibition of MCT1, with decreased GHB plasma concentrations, increased renal and total clearance, and improvements in respiratory depression in rats [28,29]. 

The objectives of the present work are to characterize the effect of exogenous testosterone treatment on GHB toxicokinetics and toxicodynamics, and to evaluate the use of the MCT1 specific inhibitor, AR-C 155858, as a potential treatment strategy for GHB overdoses in cis- and transgender males. In the present study, toxicokinetics and toxicodynamics were evaluated following 1000 mg/kg and 1500 mg/kg intravenous GHB administration in intact males, intact females (estrus), testosterone, and its corresponding placebo-treated OVX and CST rats. 

## 2. Materials and Methods

### 2.1. Chemicals

Sodium GHB was provided by the NIDA Drug Supply Program. Deuterated-GHB (GHB-d6) was purchased from Cerilliant Corporation (Round Rock, TX, USA). Formic acid was purchased from Fisher Scientific (Fair Lawn, NJ, USA). Optima™ LC/MS grade acetonitrile, methanol, and acetic acid were purchased from Fisher Scientific (Fair Lawn, NJ, USA). 6-[(3,5-Dimethyl-1H-pyrazol-4-yl)methyl]-5-[[(4S)-4-hydroxy-2-isoxazolidinyl]carbonyl]-3-methyl-1-(2-methylpropyl)thieno [2,3-d]pyrimidine-2,4(1H,3H)-dione (AR-C155858) was purchased from Chemscene (Monmouth Junction, NJ, USA). Steroids and mixed pharmaceuticals mix were purchased from Restek Corporation (Bellefonte, PA, USA). Deuterated-testosterone (testosterone-D3) was purchased from Cerilliant Corporation (Round Rock, TX, USA).

### 2.2. Animals, Estrus Staging, and Hormone Treatment

Male and female Sprague Dawley (SD) rats, ovariectomized (OVX) females, and castrated (CST) males were obtained from Charles River (Wilmington, MA, USA) at 8 weeks of age. The animals were housed under controlled temperature and humidity with an artificial 12 h light/dark cycle, and food and water were available ad libitum. The animals were acclimatized for at least one week before experiments. Hormone treatment was conducted by implanting OVX female and CST male rats subcutaneously with 60-day release pellets containing 7.5 mg testosterone or corresponding placebo pellets (Innovative Research of America) at 10 weeks of age. After 21 days of hormone exposure, rats were utilized in toxicokinetic studies. Age-matched male and female rats were used as control groups. Female rats in the estrus stage were selected for the study. Estrus cycle stages of female rats were monitored daily by vaginal lavage smear, as described in previously published literature [20,21,30,31]. The Institutional Animal Care and Use Committee (IACUC) at the University of the Pacific approved all experiments.

### 2.3. Toxicokinetic Study

Jugular vein cannulas were surgically implanted under anesthesia with ketamine/xylazine. Cannulas were flushed daily with heparinized saline to maintain patency. Rats were allowed a minimum of 72 h for recovery from surgery before the experiments were conducted. GHB was administered intravenously as a 1000 mg/kg i.v. bolus or 1500 mg/kg iv bolus with or without AR-C155858 (1 mg/kg i.v. bolus) via a jugular vein cannula to the animals at 13 weeks of age (N = 5–7 per group). Rats were dosed in metabolic cages with ad libitum access to water to allow for the concurrent collection of blood and urine samples for up to 8 h. GHB was administered as a 300 mg/mL solution in sterile water, and the time of administration was considered as time 0. AR-C155858 was administered as a 1 mg/mL solution in 10% cyclodextrin in normal saline 5 min after GHB administration. Blood samples (200 μL) were taken from the jugular vein cannula at different time points: 5, 10, 20, 30, 60, 90, 120, 180, 240, 270, 300, and 330 min for 1000 mg/kg and 10, 30, 60, 90, 120, 180, 240, 270, 300, 330, 360, and 390/480 min for 1500 mg/kg bolus with or without AR-C155858. Plasma samples were separated by centrifugation and stored at −20 °C. Urine samples were collected at 120, 240, and 360 min for 1000 mg/kg and 120, 240, and 390/480 min for 1500 mg/kg bolus with or without AR-C155858 post-dose; the total urine volume was determined, and the samples were stored at −20 °C.

The sedative/hypnotic effects (sleep time) were determined, as described in a previous publication [32]. The sedative/hypnotic duration effect of GHB was determined as the difference between the offset of the sedative/hypnotic effect (RRR) and the onset of the sedative/hypnotic effect (LRR).

### 2.4. Analysis of GHB in Plasma and Urine

GHB in plasma and urine were quantified using a validated liquid chromatography/tandem mass spectrometry (LC/MS/MS) assay that was described previously [21]. Briefly, for the preparation of plasma standards and quality controls (QCs), 5 μL of GHB-d6 and 5 μL of standard stock solution or QC stock solution were added to 45 μL of blank plasma. For each unknown plasma sample, 5 μL of GHB-d6 was added to 50 μL of plasma. Protein precipitation was then performed by the addition of 800 μL of acetonitrile with 0.1% formic acid, followed by centrifugation at 10,000 rpm for 20 min. 750 μL of the supernatant was transferred to a clean disposable glass tube and evaporated using a MultiVap Nitrogen Evaporator (Organomation, Berlin, MA, USA) under a nitrogen stream. Dried samples were reconstituted in 250 μL of 95:5 double-distilled water—acetonitrile with 0.1% acetic acid. For urine standards and QCs, 25 μL of blank urine, 5 μL of standard or QC stock solution, 5 μL of GHB-d6, and 465 μL of double-distilled water were mixed. For unknown urine samples, 25 μL of the sample was added to 5 μL of GHB-d6 and 470 μL of double-distilled water. The same LC/MS/MS method was used for GHB plasma and urine analyses. LC/MS/MS equipment consisted of an Agilent 1200 series UPLC consisting of an online degasser, binary pump, and autosampler (Agilent Technologies, Santa Clara, CA, USA) connected to a Agilent 6460 Triple Quad Mass Spectrometer. HPLC conditions, mass spectrometer parameters (Appendix A), and linear calibration ranges were described in a previous publication [21].

### 2.5. Quantification of Plasma Testosterone

Testosterone plasma concentrations were quantified using a validated LC-MS assay, as described previously [22]. In brief, for standard and quality controls, 5 μL of testosterone-D3 and 10 μL of standard or QC stock solution were added to 90 μL of acetonitrile. For unknown samples, 5 μL of testosterone-D3 were added to 100 μL of plasma. Protein precipitation was performed by adding 1000 μL of acetonitrile and centrifugation. An aliquot of the resulting supernatant (950 μL) was transferred to a clean disposable glass tube, dried under nitrogen gas, and reconstituted with the mobile phase. The same LC/MS/MS equipment was used for quantification, as mentioned above. HPLC conditions, mass spectrometer parameters (Appendix A), and linear calibration ranges were described in a recent publication [22].

### 2.6. Data Analysis

Noncompartmental analysis of toxicokinetic data was performed using Phoenix WinNonlin software Version 8.4 (Certara, Wilmington, DE, USA). The area under the plasma concentration–time curve was determined using the trapezoidal method, with AUC values extrapolated to time infinity (AUC). Total clearance (CL) was determined as dose/AUC. Cumulative urinary excretion of GHB (A_e_) was calculated by summing the amount of drug excreted in urine for all collection intervals. The fraction of drug eliminated in the urine (f_e_) was determined as A_e_/(Dose ∗ Body Weight). Renal clearance (CL_R_) was determined as A_e_/AUC. Metabolic, and nonrenal clearance (CL_M_) was determined as CL–CL_R_. Statistical differences were determined in GraphPad Prism 7 using one-way analysis of variance (ANOVA) with Tukey’s post hoc test or two-way ANOVA with Sidak’s multiple comparisons test with *p* < 0.05 considered statistically significant.

## 3. Results

### 3.1. Plasma Testosterone Levels

Plasma concentrations of testosterone are provided in Table 1. OVX and CST groups treated with testosterone had mean plasma testosterone concentrations of 6.220 ± 2.627 ng/mL and 5.440 ± 1.955 ng/mL, respectively. These concentrations fall within the established physiological range for testosterone in male animals, as previously reported [33,34]. All animals treated with testosterone placebo displayed non-detectable testosterone plasma concentrations. Intact males had a mean testosterone concentration of 0.786 ± 0.462 ng/mL, with two intact male rats having undetectable testosterone levels, possibly attributable to the daily rhythmicity of testosterone concentration in intact male rats [35,36]. Furthermore, only one intact female showed a detectable testosterone concentration of 0.481 ng/mL, a level considered within physiological ranges and consistent with prior findings [37,38].

### 3.2. Toxicokinetic Analysis

Plasma concentration time profiles following 1000 and 1500 mg/kg of GHB are displayed in Figure 1. Both OVX and CST animals treated with testosterone or placebo demonstrated non-linear GHB toxicokinetic profiles following the IV administration of 1000 mg/kg and 1500 mg/kg, which is consistent with the non-linear profiles in intact animals. AR-C155858 administration with 1500 mg/kg of GHB resulted in significantly lower GHB plasma concentrations when compared to GHB administration alone, as shown in Figure 2.

Non-compartmental analysis was conducted in Phoenix WinNonLin and the AUC, and total clearance results are presented in Figure 3 and Figure 4. Significant differences were observed in AUC between sexes, testosterone- or placebo-treated animals following i.v. administration of 1000 mg/kg (Figure 3A: *p* = 0.0157; Figure 3D: *p* = 0.0018) or 1500 mg/kg for hormone-treated CST animals only (Figure 3B: *p* = 0.0201). Consistent with AUC, total clearance varied significantly among groups at 1000 mg/kg (Figure 4A: *p* = 0.0030; Figure 4D: *p* = 0.0007) and 1500 mg/kg for the CST group only (Figure 4B: *p* = 0.0463). At 1000 mg/kg, intact females showed significantly lower AUC and higher total clearance compared to intact males and the CST testosterone placebo group (Figure 3A and Figure 4A). OVX animals treated with testosterone showed significantly lower AUC and higher total clearance compared to intact males (Figure 3D and Figure 4D). When dosed at 1500 mg/kg, the treatment of testosterone in CST animals led to significantly higher AUC when compared to the corresponding placebo group, as shown in Figure 3B. The total clearance of the CST testosterone group trended lower when compared with placebo-treated and intact animals; however, no significance was observed (Figure 4B). No significant differences were observed in AUC or total clearance in OVX groups following 1500 mg/kg of GHB administration or any of the groups administered with GHB and AR-C155858 in combination. Treatment with AR-C155858 resulted in lower AUC and higher total clearance in all animal groups, among which significantly decreased AUC was observed for testosterone-treated and intact groups (Appendix A). Concurrently, a significant increase in total clearance was observed across all groups (Appendix A).

Significant alterations in renal clearance were observed exclusively within the CST groups following administration of 1500 mg/kg of GHB (Figure 5B: *p* = 0.0020). Testosterone-treated CST males showed significantly lower renal clearance when compared with both intact groups. As the GHB dose increased, a concurrent rise in renal clearance was observed for both intact groups and the OVX testosterone placebo group, consistent with the saturation of renal reabsorption. The renal clearance of both the CST testosterone placebo group and the OVX testosterone group exhibited no discernible changes, whereas the CST testosterone group showed a noticeable reduction of renal clearance with escalating dosage. Compared to GHB administration alone, AR-C 155858 treatment led to a statistically significant increase in renal clearance across all animal groups, except for intact males (Appendix A).

The fraction of drug elimination in urine (fe) varied significantly following 1000 mg/kg for the CST group only, and at 1500 mg/kg of GHB for both animal cohorts (Figure 6A: *p* = 0.0002; Figure 6B: *p* < 0.0001; Figure 6E: *p* < 0.0001). Following 1000 mg/kg of GHB iv, placebo-treated CST animals exhibited significantly higher fe compared to both intact groups, while the CST testosterone group showed significantly higher fe when compared to intact females only (Figure 6A). As illustrated in Figure 6B, there was a significant reduction in fe in CST animals treated with either testosterone or placebo compared to intact groups at the dosage of 1500 mg/kg. Additionally, a statistically significant reduction in fe was observed in CST animals subjected to testosterone treatment when contrasted with those receiving placebo treatment. Similarly, OVX animals with testosterone treatment showed significantly lower fe when compared to both intact groups and the placebo-treated group. The OVX animals subjected to testosterone placebo treatment demonstrated significantly decreased fe in comparison to intact males (Figure 6E). No significant differences of fe were observed within OVX groups followed by 1000 mg/kg of GHB administration, as well as in the animal groups following treatment with AR-C 155858. AR-C 155858 treatment significantly increased fe in testosterone-treated animal groups compared to GHB administration alone (Appendix A).

Metabolic clearance varied significantly at both doses (Figure 7A: *p* < 0.0001; Figure 7B: *p* < 0.0001; Figure 7D: *p* = 0.0003; Figure 7E: *p* < 0.0001). Upon administration of 1000 mg/kg of GHB, intact females exhibited significantly elevated metabolic clearance in comparison to intact males, as well as the CST and OVX testosterone placebo groups and the CST testosterone group (Figure 7A). The OVX testosterone group demonstrated significantly higher metabolic clearance than that of intact males (Figure 7D). At 1500 mg/kg, OVX and CST testosterone groups, along with the CST testosterone placebo group, displayed significantly higher metabolic clearance when compared to both intact groups (Figure 7B,E). Moreover, the metabolic clearance of the OVX testosterone placebo group was significantly higher than that of intact males. As the dose increased, the metabolic clearance of intact groups exhibited a pronounced decrease, whereas the groups treated with hormones or placebos demonstrated minimal alterations. The administration of AR-C 155858 did not yield any statistically significant differences in metabolic clearance across all animal groups, except in intact males, demonstrating a significant increase in metabolic clearance following AR-C 155858 treatment compared to GHB administration alone (Appendix A).

### 3.3. Toxicodynamic Analysis

The toxicodynamics of GHB were evaluated with the sedative/hypnotic effect as the endpoint, as depicted in Figure 8. Intact males exhibited a significantly prolonged sleep time compared to the OVX testosterone group, when administered 1000 mg/kg of GHB. After 1500 mg/kg of GHB, the CST testosterone group demonstrated a significantly increased sleep time relative to the corresponding placebo group and both intact groups. The co-administration of GHB and AR-C 155858 induced a significant reduction in sleep time across all animal groups (Appendix A).

## 4. Discussion

GHB is a widely abused recreational drug, and there has been a discernible increase in its use, particularly within the LGBTQ community over the years. Considering the high frequency of ingestion of GHB within various populations, the toxicokinetic/toxicodynamic interactions need to be characterized. Furthermore, investigations into potential treatment strategies should extend to involve the prevailing conditions associated with GHB abuse, specifically within cis- and transgender populations. The current study is the first to evaluate GHB toxicokinetics/toxicodynamics subsequent to exogenous testosterone administration in CST and OVX rats representing sex and cross-sex hormone treatment. Additionally, we examined the impact of the MCT inhibitor, AR-C 155858, on the toxicokinetic/toxicodynamic profiles of GHB following sex and cross-sex hormone treatment. These findings contribute to our understanding of potential therapeutic interventions for GHB overdose across diverse demographic groups.

GHB demonstrates nonlinear toxicokinetics; testosterone- or placebo-treated CST and OVX rats had consistent PK profiles with previous studies [21,28]. At 1000 mg/kg, the plasma exposure of intact females was significantly decreased, and total and metabolic clearance were significantly increased in comparison with intact males, while renal clearance and fe remained similar. This is consistent with previous studies suggesting that females may be less susceptible to GHB-induced toxicity [21]. At this dose, CST animals treated with testosterone showed PK parameters similar to those of intact males, whereas testosterone-treated OVX rats exhibited parameters aligning with those observed in intact females. At 1500 mg/kg, the previously noted divergence between males and females at lower doses has diminished. Interestingly, testosterone-treated CST males exhibited significantly higher GHB plasma exposure (496 ± 97.6 mg*min/mL) compared to the placebo group (309 ± 23.8 mg*min/mL) and overall lower total clearance (3.2 ± 0.7 mL/min/kg) (Appendix A). Furthermore, the renal clearance (1.4 ± 0.6 mL/min/kg) and fe (0.44 ± 0.09) exhibited the lowest values in testosterone-treated CST males in comparison to placebo and intact groups (Appendix A). This implies a higher extent of renal reabsorption in testosterone-treated CST males, aligning with the higher expression levels of MCT1 and MCT4 identified in their kidneys [22]. In the rat kidney, testosterone treatment in CST animals displayed significantly upregulated MCT1 and MCT4 protein expression compared to intact animals [22]. One testosterone-treated CST animal was excluded from the analysis due to its weakened capacity for GHB elimination, leading to approximately 50% extrapolation of the AUC, which failed to show the true terminal phase and does not meet the criteria for NCA analysis. These observations suggest that CST populations undergoing testosterone treatment may be susceptible to an increased risk of GHB overdose. Upon administering 1500 mg/kg of GHB to testosterone-treated OVX rats, no significant alteration in plasma exposure and total clearance were observed when compared to placebo and intact groups. There was a general trend of lower renal clearance; however, statistical significance was not reached due to the large variance among individuals. Nevertheless, fe was significantly lower in testosterone-treated OVX rats than in placebo and both intact groups, mirroring observations in testosterone-treated CST rats.

The brain is the site of action for the toxicological effects of GHB, including sedation, hypothermia, respiratory depression, and fatality [39]. MCT1 is the only isoform present at the blood–brain barrier (BBB) and greatly contributes to the uptake of GHB [40,41]. The toxicodynamic endpoint in the current study, sleep time (RRR–LRR), exhibited a dose-dependent increase with increasing GHB dosage. This trend aligns with findings reported in prior publications [32]. At 1500 mg/kg, testosterone-treated CST rats exhibited significantly higher sleep time (an average of 422.5 min), which is almost twice the amount of time compared to placebo-treated and intact groups. Two of the testosterone-treated CST rats showed extremely long sleep times (~600 min). The greater sedative effect observed in testosterone-treated CST rats may be caused by higher MCT1 expression at the BBB, leading to higher brain–GHB concentrations. This testosterone effect on sleep time did not occur in testosterone-treated OVX rats. Further research is needed to investigate the expression levels of MCT1 on the BBB across diverse populations, as well as GHB brain distribution, to better elucidate the underlying mechanisms. A large variation in sleep time was observed within groups, especially in hormone-treated animals. This implies the limitation of using the sedative effect as the sole toxicodynamic endpoint. Future studies may consider exploring alternative endpoints, such as respiratory depression, to provide a more comprehensive evaluation of GHB toxicodynamics.

The efficacy of the MCT1 inhibitor as a potential antidote for GHB overdose has been extensively studied [28,29,42,43]. In the current investigation, the effect of the potent and specific MCT1 inhibitor, AR-C 155858, is studied for the first time in females and testosterone-treated CST and OVX rat models to determine its effects on GHB toxickinetics and toxicodynamics. The inhibition effect of AR-C 155858 is dose-dependent; a 1 mg/kg dosage of AR-C 155858 was selected for this study based on the observation that no further effect was noted at higher doses in comparison to the efficacy at the 1 mg/kg dosage [28]. Treatment with AR-C 155858 resulted in decreased plasma exposure of GHB, concomitant with increased fe, renal, and total clearance across all populations studied. Inhibition of MCTs resulted in GHB renal clearance persisting below the filtration rate, implying the potential involvement of additional transporters in the active renal reabsorption of GHB. The administration of AR-C 155858 was also able to reverse the sedative effect in all populations, reducing the sleep time to a range of 0–38 min. This is consistent with the previous finding that AR-C 155858 can alter the entry of GHB into the brain by inhibiting MCT1 localized on the BBB [28]. Although considerable elevation in sleep duration was observed in testosterone-treated CST rats following 1500 mg/kg of GHB, AR-C 155858 demonstrates sufficient potency to counteract the sedative effect. These findings suggest that this inhibitor potentially serves as an effective candidate targeting overdose treatment, despite the influence of testosterone on GHB toxicokinetics and toxicodynamics. Moreover, MCT1 inhibition via the concurrent treatment of AR-C 155858 demonstrated the ability to rectify discernible differences in toxicokinetics observed among animal groups exposed to high doses of GHB. This further supports that variations in MCT1 expression levels across the diverse populations in this study may be the main contributor to the significances we observed in GHB toxicokinetics and toxicodynamics.

When comparing 1000 mg/kg and 1500 mg/kg of intravenous GHB administration, increasing renal clearance was observed with an elevated dosage of GHB administered in the OVX testosterone placebo group and both intact groups, which is consistent with prior publications [21,28]. Renal clearance did not increase in the CST testosterone placebo group, which may be a result of the lower expression of MCTs or other GHB-related transporters compared to testosterone placebo-treated OVX animals and intact animals. The treatment of testosterone on CST animals resulted in a reduction of renal clearance from an initial average of 3.2 ± 0.9 mL/min/kg to 1.4 ± 0.6 mL/min/kg, demonstrating a dose-dependent decrease of renal clearance (Appendix A). However, increases in dose barely affected metabolic clearance, implying that metabolic pathways may be readily saturated at lower doses. This observation highlights the potential for reduced elimination of GHB in cisgender males, thereby contributing to greater concerns regarding toxicity. Interestingly, although metabolic clearance in testosterone-treated OVX rats did not change with increased dose, higher metabolic clearance (3.5 ± 1.1 mL/min/kg) was observed in these animals following 1500 mg/kg of GHB in comparison to other animal groups (<2.2 mL/min/kg), which may be a result of alterations in hepatic MCT expression (Appendix A). GHB is extensively metabolized via mitochondrial and cytosolic enzymes in the liver, and MCTs are mainly involved in the uptake of GHB into the liver [42,43]. MCTs exhibit bidirectional transporter functionality, with MCT1 predominantly facilitating the uptake of monocarboxylates, and MCT4 is involved in efflux of monocarboxylates [2]. To best evaluate GHB metabolic clearance, an extended hepatic clearance model would help elucidate transporter and enzyme interplay, which is a function of hepatic intrinsic active uptake clearance, passive transport clearance, active basolateral efflux clearance, and intrinsic biliary secretory and metabolic clearance [44]. Passive transport plays a negligible role in transporting GHB. The absence of significant differences in metabolic clearance following the inhibition of MCT1 (Figure 7C,F) implies uniformity in metabolic enzyme activity across all populations investigated in this study. The higher metabolic clearance observed in testosterone-treated OVX rats may be due to a potential upregulation of the hepatic MCT1 uptake transporter. Furthermore, we observed an approximately two-fold increase in fe with escalating dosages in intact animals and OVX placebo animals. In the context of GHB overdose, it is established that metabolic clearance will be saturated, and renal clearance emerges as the predominant elimination pathway [2]. Intriguingly, our findings deviate from this pattern following both sex and cross-sex testosterone treatment, as well as in placebo-treated CST males. In these groups, we noted either an unchanged or slightly decreased fe, suggesting hormone-dependent modulation of the GHB elimination pathways. Future studies should quantify expression changes in hepatic MCTs and GHB metabolizing enzymes following sex and cross-sex hormone treatment with testosterone to evaluate the enzyme/transporter interplay in the liver.

Testosterone is able to convert into estradiol via aromatase, a member of the cytochrome P450 family, in both men and women [45]. Aromatase, a naturally occurring enzyme, is distributed across various tissues in the body, including the brain, muscles, and testicles. In females, it is additionally present in key reproductive organs such as the ovaries and placenta [46]. In the context of normal physiological conditions in adult males and females, the conversion efficiency of testosterone to estradiol is observed to be higher in males compared to females [45]. With increasing dose of testosterone administration in young and older men, the level of estradiol results in a concomitant elevation following saturable Michaelis–Menten kinetics [47]. In contrast, estradiol concentrations persist within the physiological range for males in transgender men undergoing exogenous testosterone treatment [48]. This implies that testosterone-treated CST rats may have raised estradiol levels while testosterone-treated OVX rats may not have increased estradiol. We have demonstrated the impact of female sex hormones on GHB toxicokinetics and renal clearances [21]; however, the role of estradiol in GHB toxicity within cis- and transgender populations remains unexplored. Future investigations are necessary to elucidate the complex synergic effect of testosterone and estrogen on MCT expression and GHB metabolism enzymes to better understand GHB toxicity in cis- and transgender males. Testosterone and estrogen are known to mediate their main effects via their nuclear receptors [49]. These receptors can be activated by steroid hormones and act as ligand-regulated transcription factors that regulate a large number of target genes [50,51]. Future studies should elucidate whether sex hormone receptors are directly involved in MCT regulation.

In summary, we have observed significant alterations of GHB toxicokinetics and toxicodynamics in testosterone-treated CST and OVX rats. Our results indicate that testosterone-treated CST rats are more susceptible to GHB-induced toxicity due to increased systemic exposure and decreased renal clearance. This investigation highlights that GHB overdose among testosterone-treated men may lead to heightened severity of GHB-related adverse effects, and such outcome could potentially be attributed to variations in MCT protein expression driven by sex hormones. Additionally, future studies should evaluate the potential for GHB drug interactions with other medications prescribed during gender transition. AR-C 155858 emerges as a promising therapeutic intervention for GHB overdose in both cis- and transgender male populations. Further studies should assess sex hormone-dependent regulation of enzymes implicated in GHB metabolism. Further investigations are needed to elucidate the specific role played by female sex hormones in influencing both the toxicokinetics and toxicodynamics of GHB.

## Figures and Tables

**Figure 1 pharmaceutics-16-00143-f001:**
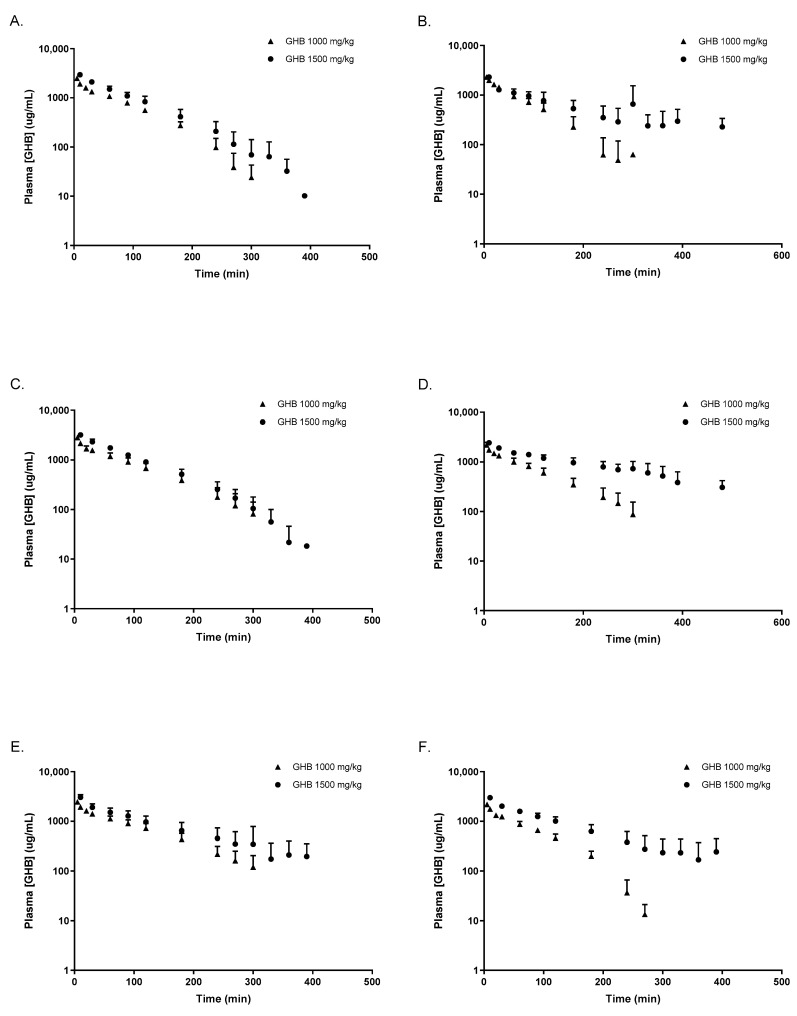
Intravenous GHB toxicokinetics following 1000 mg/kg and 1500 mg/kg of GHB i.v. (**A**) OVX testosterone placebo; (**B**) OVX testosterone; (**C**) CST testosterone placebo; (**D**) CST testosterone; (**E**) intact males; (**F**) intact females (estrus). Data are presented as mean ± SD, N = 5–7.

**Figure 2 pharmaceutics-16-00143-f002:**
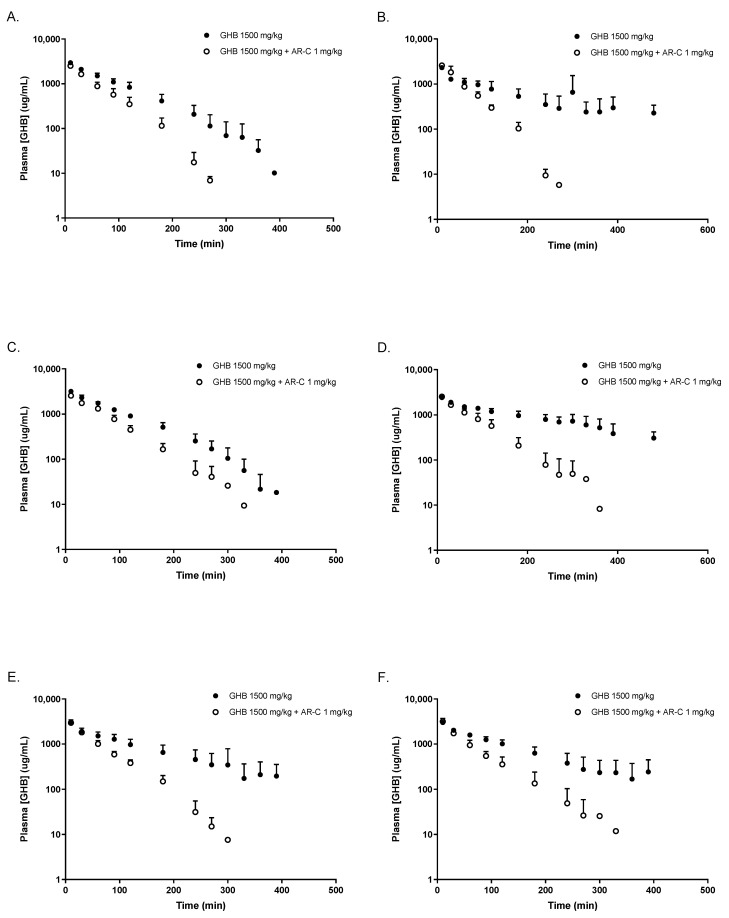
Effect of AR-C 155858 on GHB toxicokinetics following 1500 mg/kg i.v. (**A**) OVX testosterone placebo; (**B**) OVX testosterone; (**C**) CST testosterone placebo; (**D**) CST testosterone; (**E**) intact males; (**F**) intact females (estrus). AR-C 155858 was administered 5 min after GHB administration. Data are presented as mean ± SD, N = 4–6.

**Figure 3 pharmaceutics-16-00143-f003:**
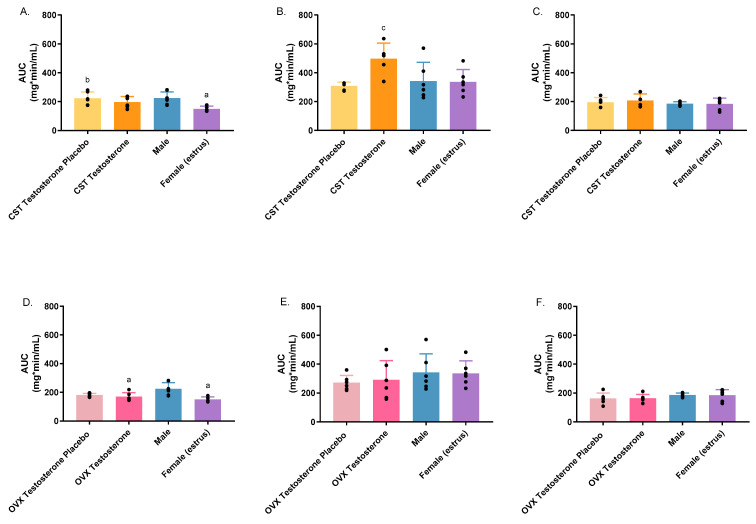
Area under the curve (AUC) following intravenous GHB administration of 1000 mg/kg (**A**,**D**); 1500 mg/kg (**B**,**E**); 1500 mg/kg + 1 mg/kg AR-C 155858 (**C**,**F**) in testosterone- and placebo-treated OVX and CST rats. ^a^
*p* < 0.05 compared with males; ^b^
*p* < 0.05 compared with females (estrus), ^c^
*p* < 0.05 compared with CST testosterone placebo group. Data are presented as mean ± SD with individual data points, N = 4–7.

**Figure 4 pharmaceutics-16-00143-f004:**
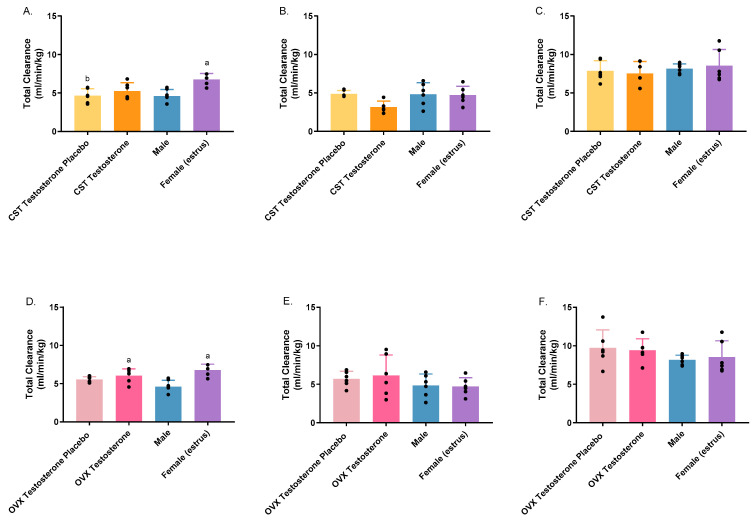
Total clearance following intravenous GHB administration of 1000 mg/kg (**A**,**D**); 1500 mg/kg (**B**,**E**); 1500 mg/kg + 1 mg/kg AR-C 155858 (**C**,**F**) in testosterone- and placebo-treated OVX and CST rats. ^a^
*p* < 0.05 compared with males; ^b^
*p* < 0.05 compared with females (estrus). Data are presented as mean ± SD with individual data points, N = 4–7.

**Figure 5 pharmaceutics-16-00143-f005:**
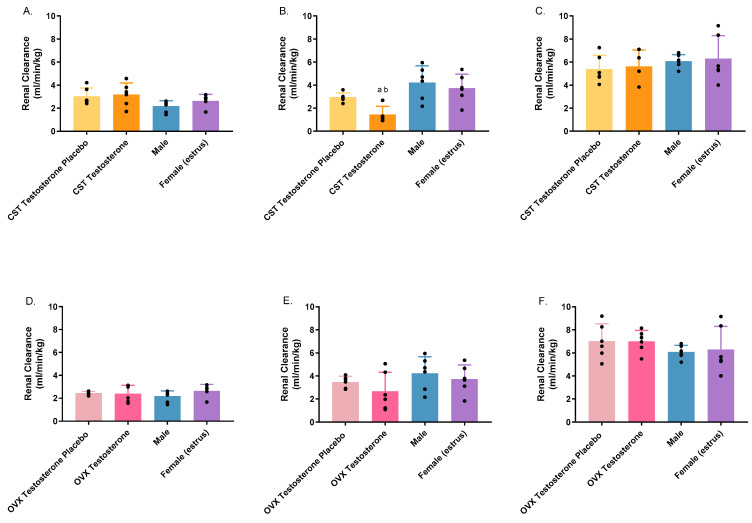
Renal clearance following intravenous GHB administration of 1000 mg/kg (**A**,**D**); 1500 mg/kg (**B**,**E**); 1500 mg/kg + 1 mg/kg AR-C 155858 (**C**,**F**) in testosterone- and placebo-treated OVX and CST rats. ^a^
*p* < 0.05 compared with males; ^b^
*p* < 0.05 compared with females (estrus). Data are presented as mean ± SD with individual data points, N = 4–7.

**Figure 6 pharmaceutics-16-00143-f006:**
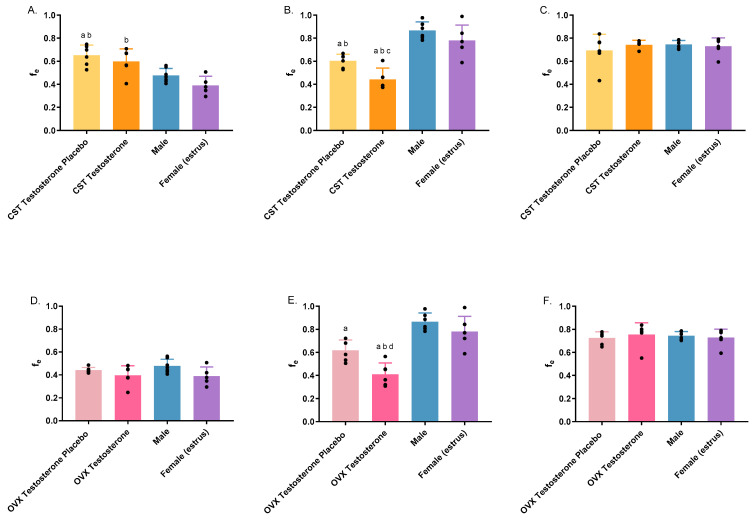
The fraction of the drug eliminated in urine (fe) following intravenous administration of 1000 mg/kg (**A**,**D**); 1500 mg/kg (**B**,**E**); 1500 mg/kg + 1 mg/kg AR-C 155858 (**C**,**F**) in testosterone- and placebo-treated OVX and CST rats. ^a^
*p* < 0.05 compared with males; ^b^
*p* < 0.05 compared with females (estrus), ^c^
*p* < 0.05 compared with the CST testosterone placebo group; ^d^
*p* < 0.05 compared with the OVX testosterone placebo group. Data are presented as mean ± SD with individual data points, N = 4–7.

**Figure 7 pharmaceutics-16-00143-f007:**
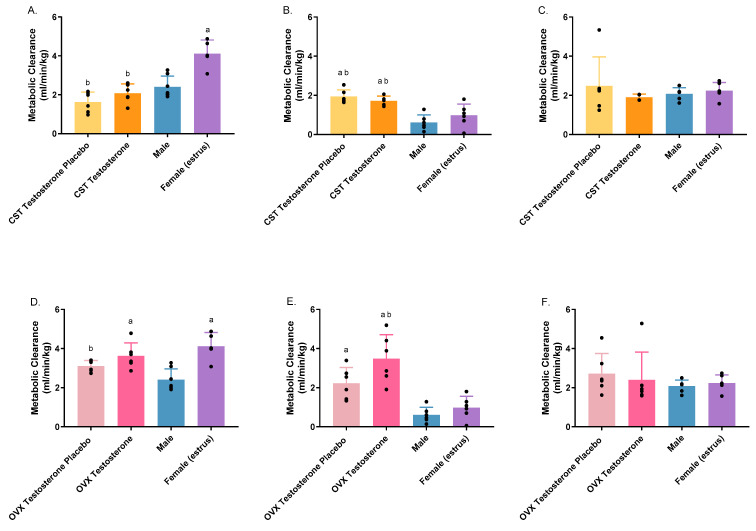
Metabolic clearance following intravenous GHB administration of 1000 mg/kg (**A**,**D**); 1500 mg/kg (**B**,**E**); 1500 mg/kg + 1 mg/kg AR-C 155858 (**C**,**F**) in testosterone- and placebo-treated OVX and CST rats. ^a^
*p* < 0.05 compared with males; ^b^
*p* < 0.05 compared with females (estrus). Data are presented as mean ± SD with individual data points, N = 4–7.

**Figure 8 pharmaceutics-16-00143-f008:**
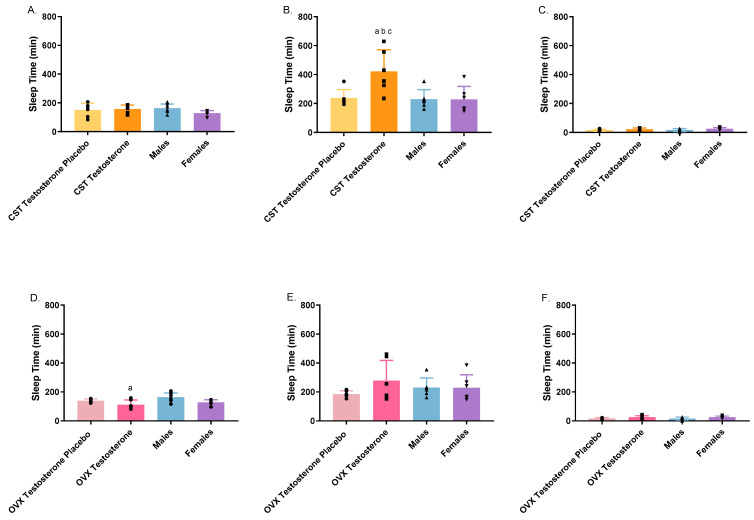
Sedative/hypnotic effect following intravenous GHB administration of 1000 mg/kg (**A**,**D**); 1500 mg/kg (**B**,**E**); 1500 mg/kg + 1 mg/kg AR-C 155858 (**C**,**F**) in testosterone- and placebo-treated OVX and CST rats. Sleep time represents the differences between RRR and LRR. ^a^
*p* < 0.05 compared with males; ^b^
*p* < 0.05 compared with females (estrus); ^c^
*p* < 0.05 compared with the CST testosterone placebo group. Data are presented as mean ± SD with individual data points, N = 5–8.

**Table 1 pharmaceutics-16-00143-t001:** Plasma testosterone levels in intact females, males, testosterone, or its corresponding placebo-treated OVX and CST rats.

Plasma Hormone Level	Testosterone (ng/mL)	Testosterone Non-Detected
OVX Testosterone Placebo	--	18 out of 18
OVX Testosterone	6.220 ± 2.627	0 out of 18
CST Testosterone Placebo	--	18 out of 18
CST Testosterone	5.440 ± 1.955	0 out of 17
Females (estrus)	0.481	16 out of 17
Males	0.786 ± 0.462	2 out of 21

## Data Availability

The data presented in this study are available within the article and Appendix A.

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
