# Peer review of "Sex and Cross-Sex Testosterone Treatment Alters Gamma-Hydroxybutyrate Acid Toxicokinetics and Toxicodynamics in Rats"

_pharmaceutics, 2024, doi:10.3390/pharmaceutics16010143_

Round 1

Reviewer 1 Report

Comments and Suggestions for Authors

This study investigates the impact of administrations of teststerone and MCT1 inhibitor on GHB toxicokinetics and toxicodynamics in OVX female rats and castrated (CST) male rats. This study is interesting, but the review has the following criticisms.

1. Correction for variation in pathogens is an important issue for quantitative analysis of toxicokinetics and toxicodynamics. What internal standards were used?

2. There is no pharmacokinetic analysis of plasma concentration. The discussion should be quantitatively based on the values of estimated parameters (systemic clearance, volume of distribution, etc.).

Author Response

This study investigates the impact of administrations of teststerone and MCT1 inhibitor on GHB toxicokinetics and toxicodynamics in OVX female rats and castrated (CST) male rats. This study is interesting, but the review has the following criticisms.

  1. Correction for variation in pathogens is an important issue for quantitative analysis of toxicokinetics and toxicodynamics. What internal standards were used?

For quantitative LCMS analysis, GHB-d6 and testosterone-D3 were used as internal standards for GHB and testosterone, respectively (detailed in Section 2.4 and 2.5).

  1. There is no pharmacokinetic analysis of plasma concentration. The discussion should be quantitatively based on the values of estimated parameters (systemic clearance, volume of distribution, etc.).

The quantitative data of pharmacokinetic analysis were addressed in the supplementary information Table S3, S4 and S5. Data analysis is described in Section 2.6

Reviewer 2 Report

Comments and Suggestions for Authors

The authors performed a great trial to evaluate variability in the toxicokinetics of GHB. I would advise the authors to review the discussion and to point more extrapolation with possible interaction in medication used for gender transition.

Author Response

The authors performed a great trial to evaluate variability in the toxicokinetics of GHB. I would advise the authors to review the discussion and to point more extrapolation with possible interaction in medication used for gender transition.

Statements and references were added (line 468-471, 478 – 80).

Reviewer 3 Report

Comments and Suggestions for Authors

Zhang et al. performed a study on the impact of sex and cross-sex testosterone treatment on GHB toxicokinetic and toxicodynamic in rats. The author has shown how the exogenous testosterone treatment induces significant alterations in GHB toxicokinetic and toxicodynamic, and MCT inhibition can serve as a potential therapeutic strategy for GHB overdose in both cisgender and transgender male populations. Please, find below my comments.

1.       The sex differences in rodents are more prominent than in humans. Please, cite more references where the MCAT shows differences in the expression male vs female in humans.

2.       Also, please highlight in the discussion how this finding will be translated into humans.

3.       Please, cite enough evidence of GHB toxicity in humans, if reported.

4.       How was the TK dose selected? Please explain.

5.       What is the impact of testosterone on the expression of MCAT transporter? If there is any literature evidence, please cite the reference.

6.       Though the author mentioned in the manuscript validated LC-MS assay, I don’t see any method validation parameters reported in the manuscript. Please, include commonly used LC-MS method validation parameters in the supplementary information.

7.       Does the author measure GHB concentration in the brain at a terminal time point? If so, please report and compare with the control versus treatment arm.

8.       Does the author detect any changes in GHB overdose toxicity at the cellular level?

9.       Please make the referencing style consistent. 

Author Response

Zhang et al. performed a study on the impact of sex and cross-sex testosterone treatment on GHB toxicokinetic and toxicodynamic in rats. The author has shown how the exogenous testosterone treatment induces significant alterations in GHB toxicokinetic and toxicodynamic, and MCT inhibition can serve as a potential therapeutic strategy for GHB overdose in both cisgender and transgender male populations. Please, find below my comments.

  1. The sex differences in rodents are more prominent than in humans. Please, cite more references where the MCAT shows differences in the expression male vs female in humans.

Statements and references were added.

  1. Also, please highlight in the discussion how this finding will be translated into humans.

Statements were added.

  1. Please, cite enough evidence of GHB toxicity in humans, if reported.

Statements and references were added to the introduction section.

  1. How was the TK dose selected? Please explain.

As reported in the literature, the LD50 dose of GHB is 1750 mg/kg. Previous studies demonstrated that 1500 mg/kg GHB was the maximal dose that could be administered without causing death in adult male rats. As GHB demonstrates nonlinear toxicokinetics two doses were selected to evaluate the nonlinearity in toxicokinetic parameters. The doses of 1000 mg/kg and 1500 mg/kg were selected to be consistent with previous studies conducted in adult male rats.

  1. What is the impact of testosterone on the expression of MCAT transporter? If there is any literature evidence, please cite the reference.

We have previous evaluated renal MCT/SMCT expression following testosterone treatment (Wei et al. 2023 – ref 22). Additionally, testosterone treatment has been demonstrated to regulate muscle MCT expression (Ref 17).

  1. Though the author mentioned in the manuscript validated LC-MS assay, I don’t see any method validation parameters reported in the manuscript. Please, include commonly used LC-MS method validation parameters in the supplementary information.

Table S1 and S2 were added and contain the assay validation. A reference is included in the methods section to the original publication.

  1. Does the author measure GHB concentration in the brain at a terminal time point? If so, please report and compare with the control versus treatment arm.

In the present study, GHB brain concentrations at the end of the 8 hour collection would be minimal based on previously published brain microdialysis data. This collection interval was necessary to determine the terminal slope and renal excretion/clearance of GHB. Future studies are planned to investigate the brain GHB concentration at a terminal time points in different populations.

  1. Does the author detect any changes in GHB overdose toxicity at the cellular level?

We did not perform any experiments at cellular level.

  1. Please make the referencing style consistent.

References revised.

Round 2

Reviewer 3 Report

Comments and Suggestions for Authors

I have no further comments.